# Upregulated Chemokine and Rho-GTPase Genes Define Immune Cell Emigration into Salivary Glands of Sjögren’s Syndrome-Susceptible C57BL/6.NOD-*Aec1Aec2* Mice

**DOI:** 10.3390/ijms22137176

**Published:** 2021-07-02

**Authors:** Ammon B. Peck, Cuong Q. Nguyen, Julian L. Ambrus

**Affiliations:** 1Department of Infectious Diseases and Immunology, College of Veterinary Medicine, University of Florida, P.O. Box 100125, Gainesville, FL 32610, USA; nguyen@ufl.edu; 2Division of Allergy, Immunology and Rheumatology, SUNY Buffalo School of Medicine, 875 Ellicott Street, Buffalo, NY 14203, USA; jambrus@buffalo.edu

**Keywords:** Sjögren’s syndrome, marginal zone B cells, RNA transcriptome microarray, Rho-GTPases, GTP-GAP, GTP-GEF, DOCK molecules, C57BL/6.NOD-*Aec1Aec2* mice

## Abstract

The C57BL/6.NOD-*Aec1Aec2* mouse is considered a highly appropriate model of Sjögren’s Syndrome (SS), a human systemic autoimmune disease characterized primarily as the loss of lacrimal and salivary gland functions. This mouse model, as well as other mouse models of SS, have shown that B lymphocytes are essential for the development and onset of observed clinical manifestations. More recently, studies carried out in the C57BL/6.*IL14α* transgenic mouse have indicated that the marginal zone B (MZB) cell population is responsible for development of SS disease, reflecting recent observations that MZB cells are present in the salivary glands of SS patients and most likely initiate the subsequent loss of exocrine functions. Although MZB cells are difficult to study in vivo and in vitro, we have carried out an *ex vivo* investigation that uses temporal global RNA transcriptomic analyses to profile differentially expressed genes known to be associated with cell migration. Results indicate a temporal upregulation of specific chemokine, chemokine receptor, and Rho-GTPase genes in the salivary glands of C57BL/6.NOD-*Aec1Aec2* mice that correlate with the early appearance of periductal lymphocyte infiltrations. Using the power of transcriptomic analyses to better define the genetic profile of lymphocytic emigration into the salivary glands of SS mice, new insights into the underlying mechanisms of SS disease development and onset begin to come into focus, thereby establishing a foundation for further in-depth and novel investigations of the covert and early overt phases of SS disease at the cellular level.

## 1. Introduction

Autoimmunity is generally recognized as a multi-step process initiated by environmental triggers that first activate an innate inflammatory reaction, then subsequently progress to an adaptive immune response, in genetically or physiologically predisposed hosts, that targets critical biological functions whose subsequent dysfunctions eventually result in an overt clinical pathology [1,2,3,4,5,6]. As such, autoimmune diseases are considered to have both an early *covert* disease phase and a late *overt* disease phase. Unfortunately, patients most often present in clinic only after the adaptive immune phase is active and irreversible pathology is occurring. For this reason, it remains necessary to identify molecular and cellular processes involved with the early covert inflammatory phase of autoimmunity as a basis to better understand the initiation of the adaptive immune response phase. This is especially true for Sjögren’s Syndrome (SS) where the time between disease onset and eventual diagnosis ranges from 4 to 10 or more years [7,8,9]. Furthermore, the pathological profile changes dramatically during the development and onset of disease in this time period.

SS is a highly debilitating, yet fascinating systemic autoimmune disease marked by leukocytic infiltrations of the salivary and lacrimal glands with a concomitant loss of exocrine secretion leading to clinical symptoms of severe dry mouth and dry eye diseases (reviewed in [9]). Occasionally, patients will develop kidney and lung infiltrations, central nervous system manifestations, and even lymphomas, the latter most commonly non-Hodgkin’s B cell lymphomas [6]. While considerable effort has long focused on defining the general pathophysiology and accompanying symptoms for disease diagnosis in patients, only recently has a more concerted effort surfaced to better define the molecular-based bioprocesses for the development and subsequent onset of SS autoimmunity and lymphomagenesis, research that has mostly been carried out utilizing a wide variety of mouse models [10,11,12].

Over the past couple decades, we have investigated multiple mouse models that exhibit progressive and spontaneous SS-like pathologies to define the early molecular events responsible for disease development and onset [13,14,15,16,17,18,19,20,21]. In general, these studies have defined this SS-like disease development as: (a) aberrant proteolytic enzyme activity most likely associated with acinar cell apoptosis, (b) a progressive loss of saliva and tear flow rates accompanied by increased protein content due in part to less fluid secretions, (c) decline in amylase and carbonic anhydrase activities, (d) appearance of autoantibodies, and (e) loss of acinar tissue, all manifestations occurring concomitantly with increasing glandular lymphocytic infiltrates. More recent studies have now shown a direct correlation between an upregulated expression of IL14α and late-stage B cell lymphomagenesis [13,14]. Importantly, global temporal transcriptome studies have verified these reported immuno-pathophysiological temporal changes, as well as the fact that full development of disease is clearly a multi-phase process involving an innate response, an adaptive response, and in some individuals, a late lymphomagenesis phase. This progression of autoimmune pathologies is quite significant when compared against sex- and aged-matched SS-non-susceptible (SS^NS^) control mice that do not develop clinical SS-like diseases.

While earlier studies revealed that inactivating the B cell receptor [14,15], interferon [16], IL-4 (interleukin-14) [17] and/or the alternate C’ system [18] prevents SS-like disease in our mouse models, our recent studies with B6.*Il14α* transgenic (TG) mice [19,20] have now indicated that elimination of marginal zone B (MZB) cells, or the blocking of lymphotoxin activity that is required for MZB cell ontogeny in marginal zones (MZ), prevents the development of SS-like disease, including lymphomagenesis [21,22]. MZB cells are an unique subpopulation of bone marrow derived B cells characterized by limited expression of immunoglobulin variable region genes that produce predominantly IgM antibodies, many of which are self-reactive [23]. In addition, they are strategically located within mucosal surfaces, function as innate and/or transitional cells capable of rapid responses to both T cell-independent and T cell-dependent antigens, and regulate the activation of subsequent adaptive immune responses by T and B2 lymphocytes in association with monocytic and neutrophilic antigen-presenting cells (APCs) (reviewed in [23]). Functionally, MZB cells differentiate from transitional type-1 (T1) B cells under the influence of low affinity B cell receptor (BCR) signals and transcription factors, especially Notch-2 [24]. They are enriched within splenic MZs, retained there by interactions between MZ integrin and MZB cell integrin receptors (e.g., MADCAM1 (mucosal cell adhesion molecule-1), LFA1 (lymphocyte function-association-1), ICAM (intercellular adhesion molecule-1), VLA-4 (very late antigen-4) and SIP1 (sphingosine-1-phosphate) [25]. MZB cells respond rapidly to Toll-like receptor (TLR) signaling, especially TLR2, 4 and 9 [25,26]. While MZB cells are important for responding to pathogens [27,28], they have been identified within the salivary glands of patients with salivary gland disease where they secrete cytokines cytotoxic to salivary gland cells [22,29,30].

Despite the noted difficulty in studying MZB cells either in vivo or in vitro, we have utilized *ex vivo* temporal global transcriptomic microarrays to identify MZB cell markers in order to better define MZB cell profiles and their temporal presence in SS-like disease of the SS-susceptible (SS^S^) C57BL/6.NOD-*Aec1Aec2* mouse model. In the present report, we have analyzed global microarray data to identify the temporal genetic profile associated with the emigration of cells, predominantly MZB cells, into the salivary glands during early stages of SS-like disease development and onset in C57BL/6.NOD-*Aec1Aec2* mice.

## 2. Results

### 2.1. Salivary Gland Histology Pre- and Post-Lymphocyte Infiltration in C57BL/6.Aec1Aec2 Mice

One of the main manifestations of SS/SS-like disease is the progressive and age-dependent leukocytic infiltrations of the salivary and lacrimal glands, resulting in the progressive formation of lymphocytic foci (comparison of Figure 1A to Figure 1B). While this emigration of B and T cells is considered an important bioprocess and is easily visualized histologically, little is known about its molecular basis. However, results from our earlier studies using several SS-susceptible mouse models revealed an absolute requirement for B lymphocytes in order for an overt clinical SS-like disease to develop, irrespective of whether T lymphocytes infiltrate the salivary or lacrimal glands [12]. Furthermore, multiple studies have now suggested that MZB cells are the B cell population responsible for disease development [22,31], a concept not only supported by the studies in B6.*Il14α* TG mice discussed above, but also by our recent findings in C57BL/6.NOD-*Aec1Aec2* mice of a coordinated activation of the Notch2 and Type 1 interferon pathways in the absence of an upregulated Notch1 pathway [32]. This profile, in turn, predisposes mice to a prolonged upregulated Type1 interferon response characteristic of SS disease [33], our hypothesized anti-dsRNA reaction underlying SS-like disease [34], and induction of high expression of C´ (complement) receptors for the alternate C’ activity [35,36].

### 2.2. Gene Expression Profiles Defining, in Part, the Early Leukocyte Emigrations into the Salivary Glands of C57BL/6.NOD-Aec1Aec2 Mice

#### 2.2.1. Coordinated Upregulation of Chemokine > Chemokine Receptor Genes with LFA1 and Rho-GTPase Upregulations

An interesting phenomenon that distinguishes MZB cells from follicular B cells, particularly in mice, is the relatively non-circulatory state of MZB cells compared to the high migratory propensity of follicular B cells [37]. This difference, as stated by Lu and Cyster [25], occurs because of the upregulated surface levels of integrin expression on MZB cells, especially LFA-1. LFA-1 is a heterodimeric integrin comprised of subunits ItgαL and Itgβ2 that binds to ICAM-1 and VCAM1. Development of splenic MZ areas during ontogeny requires the lymphotoxin-α1:β2 integrin-mediated induction of VCAM-1 and ICAM-1, the latter a primary receptor for LFA-1. At the same time, this high level of integrin expression on MZB cells requires higher threshold levels of Cxcl13 (chemokine (CXC motif) ligand) signaling for Cxcr5 (chemokine (CXC motif) receptor) expressing MZB cells to be dislodged from MZs and emigrate to other tissue sites. As presented in Figure 2, our transcriptomic data reveal that genes encoding Icam1, Vcam1, ItgαL, Itgβ2, Cxcl13, and Cxcr5 (the Cxcl13 receptor) are all upregulated in the salivary glands at the same time (i.e., 16 weeks of age), along with Ccr6, Ccr7, and Ptk2β, strongly suggesting an influx of immune cells actively recruited by glandular tissue signaling at the early stage of disease. In addition, the development and accumulation of MZB cells in MZs, followed by their subsequent migration, appear to rest in the requirement for the *Dock2 > Rho-GEF > Ptk2* molecular pathway [28,38]. At the same time, it is important to note the published work of Gotoh et al. [39] showing that Dock2 is also indispensable for the co-migration of plasmacytoid dendritic cells (pDC), a highly relevant fact considering the extensive type 1 IFN-signature seen in exocrine glands of both human and mouse SS diseases.

#### 2.2.2. Transcriptomic Profile of the Rho-GTPase Family of Proteins

Cell emigration is highly dependent on two events: chemokine signaling to direct leukocytes to sights of injury, and Rho-GTP (Ras homolog-G protein bound-guanine nucleotide) family protein (Rho-GTPase) activations. In a quiescent state, the Rho-GTP proteins mostly localize to cellular membranes but dissociate from these membranes during initiation of migration. In mice, the Rho-GTP family consists of 21 members subdivided into several subfamilies: 11 Rho proteins, 3 Rnd proteins, 3 Rac proteins, 3 Rhobtb proteins and Cdc42 [38]. Interestingly, as shown in Figure 3, a single molecular entity from each subfamily exhibits a temporal activation (i.e., *Rhoc*, *Rhou*, *Cdc42*, *Rnd1*, and *Rhobtb1*), except for Rsf3, where both *Rac2* and *Rac3* are upregulated. These activations temporally mimic the development of SS-like pathology within the salivary glands, with *Rhoc*, *Cdc42*, *Rac3*, and *Rhou* exhibiting early upregulation starting at 8 weeks of age, and with *Rac2* and *Rhobtb1* showing an upregulation at 16 weeks, an age when the adaptive response is observed in the salivary glands of these mice.

#### 2.2.3. Transcriptomic Profile of the Rho-GAP and Rho-GEF Subfamilies of Rho-GTPase Proteins

The Rho-GTP proteins are further divided into two functional classes, those that are incapable of hydrolyzing GTP (i.e., Rhoh and Rnd subfamily members that constitutively bind GTP), and those that hydrolyze GTP through an opposing equilibrium of GTPase-activating proteins (GAPs) and Rho-specific guanine nucleotide exchange factors (GEFs) [38,39,40]. GEFs activate Rho-GTPases by exchanging a bound GDP with a GTP, while GAPs inactivate Rho GTPases *vis a vis* GTP by catalysis. Comparative temporal gene expression profiles for GAP and GEF molecules in the salivary glands of C57BL/6.NOD-*Aec1Aec2* mice versus C57BL/6J mice, summarized in Figure 4, reveal a highly restricted activation with only 7 of the 17 *Arhgef* (Rho GTPase exchange family) gene members and 8 of the 23 *Arhgap* (Rho GTPase activating family) gene members upregulated on the arrays. These gene set profiles are markedly different from the profiles seen in the salivary glands of C57BL/6J mice, where there is a marked absence of upregulated gene expressions within the same gene sets.

In addition to the Arhgef proteins, a second subset of Rho-GEF proteins are the DOCK (dedicator of cytokinesis) family proteins (reviewed in [41]). DOCK molecules, also known as CZH proteins, possess a functional CZH2 domain that promotes the exchange of GDP to GTP in the formation of Rho-GTPases. Based on sequence homologies represented within the eleven identified DOCK molecules, this molecular family of proteins is currently divided into four major subgroups, (A) Dock1, 2 and 5, (B) Dock3 and 4, (C) Dock6, 7, and 8, and (D) Dock9, 10 and 11. Their temporal gene expression profiles in SS-susceptible C57BL/6.NOD-*Aec1Aec2* mice relative to their gene expression profiles in SS-non-susceptible C57BL/6J control mice are shown in Figure 5. As might be expected in a family of proteins critical for multiple cellular functions, but especially cell migration and homeostasis, C57BL/6.NOD-*Aec1Aec2* mice exhibit a distinct transcriptomic profile involving both in activation and in temporal expressions: *Dock3*, *Dock4*, *Dock6*, and *Dock9* remain quiescent, *Dock1*, *Dock5*, *Dock7* and *Dock8* exhibit relatively long-term temporal upregulated expressions peaking between 8 and 12 weeks of age, while *Dock2*, *Do**ck10* and *Dock11* exhibit a unique short-term upregulated profile occurring around 16 weeks of age, a timepoint when the salivary glands are first being invaded by leukocytes. In contrast, SS-non-susceptible C57BL/6J mice show only four DOCK genes with upregulated expressions: *Dock4*, *Dock7*, *Dock9* and *Dock10*, and in each case maximum expressions occurred between 8 and 12 weeks of age, consistent with the occasional leukocyte infiltrations observed in the salivary glands of C57BL/6J mice.

#### 2.2.4. Transcriptomic Profiles of Factors Associated with Signal Transduction Pathways of Rho-GAP and Rho-GEF Subfamily Proteins of the Rho-GTPases

Arhgef molecules interact with several unique molecular families, including the Rac-GTPs, Rho-GTPs and cell division cycle-42 (Cdc42)-GTPs (reviewed in [42,43]). These molecules, in turn, interact with Wasf (WASP Family member 3) protein, Rock1 and/or Rock2 (Rho-associated coiled-coil containing protein kinase 1 and protein kinase 2) proteins, or Was (Wiskott–Aldrich molecule) proteins, respectively. Various functions regulated by these pathways include focal adhesions, myosin activations, lamellipodia formation, membrane ruffling, actin stabilization and polymerization, and filopodia formation, each an important factor in cytokinesis and cell migrations. Our transcriptome analysis, presented in Figure 6, reveals that the genes encoding for *Wasf3*, both *Rock1* and *Rock2*, and *Was* are each upregulated in the salivary glands of C57BL/6.NOD-*Aec1Aec2* mice. In contrast, only *Wasf3* is highly upregulated and *Rock1* minimally upregulated in the salivary glands of C57BL/6J control mice. These data point to an activation of Rho-GTP pathway(s) in C57BL/6.NOD-*Aec1Aec2* mice at 8 weeks of age, followed by an activation of Cdc42 pathway(s) at 16 weeks of age in-line with the two early disease phases for SS-like disease in the salivary glands of our SS^S^ C57BL/6.NOD-Aec1Aec2 mice.

Rho-GTPases are known to interact with multiple proteins that form signal transduction pathways that regulate activations of cellular biological functions. These include members of the Cdc42 protein family (e.g., Cdc42bpa, Cdc42bpb, Cdc42ep4, Cdc42se1 and Cdc42se2), the Vav family (e.g., Vav2, Vav3) and the Baiap family (e.g., Baiap2, Baiap2l1). In addition, genes encoding other interacting proteins include *Setd6* (*Prex1*) (Ser domain containing 6), *Scrib1* (Scribble homolog), *Flii* (Flii actin remodeling protein), *Grb2*, *Creb3* (Camp response element binding protein), and *Foxo3a*, the majority of which exhibit upregulated gene profiles in the salivary glands of C57BL/6.NOD-*Aec1Aec2* mice (Figure 7). In contrast, genes encoding *Tiam1* (Rac1 associated GEF1) and *Srgap1* (Synaptic Ras GTPase-activating protein 1), known to suppress Rho-GEFs, demonstrate no upregulated expressions. At the same time, the genes encoding *Stat6* and *Stat1* (Signal transduction and activator of transcription factors 6 and 1) show temporal upregulated expression profiles consistent with their known functional activities in SS, i.e., Stat6 being important during early-stage disease and Stat1 being important in the later stages of disease.

## 3. Discussion

The focus of the present transcriptome study has been to profile the underlying molecular bases for emigration of the autoimmune cellular infiltrates to the salivary glands during the early stages of disease development and onset of SS-like pathology observed in the C57BL/6.NOD-*Aec1Aec2* mouse model. The data presented indicate an active upregulation of signal transduction pathways defining cell immigrations involving specific GTPases. This differentially expressed gene profile occurs from 8 weeks of age (Figure 3, Figure 4, Figure 5, Figure 6 and Figure 7), a timepoint corresponding with the hypothesized appearance of MZB cells within the salivary glands. In contrast, an active chemokine expression, together with an environment for leukocyte adhesion in the salivary gland, appears to occur uniquely at 16 weeks of age (Figure 2), the time point corresponding to the infiltration of increasing numbers of leukocytes and the establishment of lymphocytic foci. Thus, we propose that this early disease stage can be divided into a ¨*passive*¨ emigration involving MZB cells and an ¨*active*¨ emigration involving recruitment of additional leukocyte populations. Nevertheless, the precise cellular functions attributed to MZB cells in SS still require better definition, since MZB cells are considered better responders to pathogens than to self-antigens [44,45], and responses toward microorganisms or other environmental triggers have yet to be ruled out as underlying initiators of SS. In this regard, our earlier transcriptome studies of the interferon-signature and activated TLR pathways point to the possibility of an anti-dsRNA virus response [11,13,14].

As an immune cell population, MZB cells show: (a) upregulated Notch2, Adam10, RBP-J and their associated molecular pathways for B cell ontogeny and differentiation [36]; (b) expressions of VLA-4 and SIP1 [45]; (c) rapid responses toward LPS while focusing T cell responses toward pathogens at restricted locations [46]; (d) strong responses to TLR2, TLR4 and TLR9 signaling [47,48]; (e) a capacity for delivering antigens to follicular dendritic cells in germinal centers [49], presentation of antigen efficiently to T cells [50], and regulation of both T cell-independent and T cell-dependent B and T cell responses [51]. Phenotypically, both mouse and human MZB cells are identified as IgM^hi^_,_ IgD^low^, CD21^hi^, and CD23^−/−^, but mouse MZB cells also express Cd1d^hi^, while human MZB cells express CD1c and CD27. Interestingly, although the microvasculature of mouse and human splenic marginal zones (MZs) are anatomically different, with human spleens apparently lacking a marginal sinus that can influence migratory pathways, both species provide an environment for MZB cells to interact with APCs in their respective stromal reticular cell networks [52], thereby permitting efficient immune surveillance of blood-borne antigens. Furthermore, both species separate MZB and follicular B cells via the periarteriolar lymphoid sheaths (PALS). Despite these important differences noted between mouse and human MZB cell ontogeny and behavior that include sites of potential maturation, resident locations, and antibody isotype repertoires (especially to T cell-dependent antigens), both mouse and human MZB cells tend to utilize similar activation signaling, e.g., BAFF, APRIL, CD40L, TLRs, IFN, IL6, IL10, and/or CXCL10 produced by APCs, to activate TACI and BCR for antibody production and immunoglobulin class switching [23].

For the present study, the overriding question is whether MZB cells function as the inducers of innate autoimmunity [53,54,55]. This would require a correct stromal cell environment within the MZ areas and, most likely, the presence of the immune-targeted autoantigen(s) to which extra-nodal lymphomagenesis can occur. Ontological development of MZs require the induced expression of ICAM1, VCAM1 and MADCAM that, in the presence of IL-7, recruits lymphoid tissue inducer cells (LTIs) that express RXRγt (retinoid X receptor gama) to sites surrounding blood vessels [56]. LTIs secrete lymphotoxin that, in turn, stimulates secretion of chemokines that attract dendritic and lymphoid cells, plus regulate endothelial cells delineating marginal sinuses [57]. Three additional stromal cell populations that dictate ontological effects on MZs are the marginal reticular cells (MRCs) which express RANKL (receptor activator of NF-*k*β ligand), fibroblast reticular cells (FRCs) which along with MRCs regulate immune responses via production of reticular fibers and chemokines that guide lymphocyte trafficking, and follicular dendritic cells (FDCs) which secrete CXCL13 and BAFF to enhance B cell maturation, survival and emigration [58]. Importantly, these stromal cell populations secrete type 1 interferon (IFN1) in response to B cell secretion of lymphotoxin, giving rise to IFN-signatures that are common to the rheumatoid autoimmune diseases. As shown here, the salivary glands of C57BL/6.NOD-*Aec1Aec2* mice exhibit upregulated *Icam1*, *Vcam1* (the two LFA1 integrin units), *Cxcl13* and chemokine receptors expressed by MZB cells, all at 16 weeks of age, the time frame when increasing numbers of lymphocytes are seen histologically entering the exocrine glands. As expected, *Madcam1*, not expressed in salivary glands, is not upregulated (data not shown).

The present analysis also reveals a distinct temporal profile for the Rho-GTPase families of proteins in the salivary glands of SS^S^ C57BL/6.NOD-*Aec1Aec2* mice as compared to SS^NS^ C57BL/6J mice. Rho-GTPases play a critical role in signal transduction pathways involving multiple biological processes, especially cell migration and tissue homeostasis. Although we previously identified the *Rac > Ras > Raf > Erk > AP* gene pathway as a critically important component in development of glandular pathology [59], the current in-depth analyses of the Rho-GTPases points to a highly complex biological process indicated by the upregulation of genes encoding for multiple Rho-Gap, Rho-Gef, Dock and Cdc42 sub-family proteins of the Rho-GTPase family. Additional molecules include proteins involved in signal transduction pathways influencing cell functions and cell migrations, such as Was, Wasf and Rock [42,43].

Rho-GEF associated molecules are known to be critical for B cell functions. For example, Tedford et al. [60] reported that Vav1/Vav2-negative B cells are unresponsive to thymus-independent antigens in vivo and indicated a role for Vav-2 in BCR calcium signaling that is critical to B cell development and function. Similarly, Wang et al. [61] have reported that membrane-proximal BCR signaling molecules (including Vav3, Lyn, Syk, Btk, PLC-γ2, and Blnk), together with adaptor molecules Grb2, Cbl and Dok3, actually link BCR micro-clusters and motor proteins. Our transcriptome data indicate that *Vav2* and *Vav3*, but not *Vav1*, exhibit upregulated activation profiles in the salivary glands of C57BL/6.NOD-*Aec1Aec2* mice, while in C57BL/6J mice, *Vav1* is strongly upregulated with *Vav2* and *Vav3* weakly upregulated. Sorting out the exact profiles for the various signal transduction pathways is the next step in better understanding these data.

In contrast, a direct correlation exists between upregulated gene expressions of Rho-GEF DOCK subfamily molecules encoding Dock2, 8, 10 and 11 with cellular homeostasis, development, and migration. Interactions of DOCK molecules with the Rho-family CDC42 proteins are also known to activate conformational changes in p21-activated PAK family molecules that regulate actin reorganization critical to cell adhesion and invasiveness, as well as to upregulate RAF and RAS pathways leading to activations of ERK, NF-kB, and AP1 (c-Jun, c-Fos and Atf2). As reported by Parrado and colleagues [62], expression of DOCK subgroup D molecules (i.e., Dock9, 10 and 11) occurs in peripheral blood lymphocytes; however, more recently, evidence has suggested that Dock10 expression is uniquely upregulated in B cells by IL-4. Furthermore, *Dock10.1* isoform regulates T cell activities, while *Dock10.2* isoform regulates CD23 expression on B cells, sustains B cell lymphopoiesis in secondary tissues, and up-regulates the IL4>Stat6 pathway [62], an essential pathway for SS development in C57BL/6.NOD-*Aec1Aec2* mice [17]. This complexity has been expanded by Fukui et al., whose studies demonstrated that *Dock2-Pi3k-delta* activation regulates B cell migration and proliferation, while *Dock2-Pi3k-gamma* activation regulates T cell migration and proliferation [63,64]. Although the importance of *Dock10* cannot be understated in B cell activation, *Dock2* gene knockout mice are deficient in MZB cells [63], as are mice with deficiencies in *Rac2*, *Ptk2* or *Rho-GEFs*.

Lastly, while *Dock2*, *10*, and *11*, as well as *LFA1*, *Cxcr5*, *Cxcl13*, and chemokine receptors *Ccr6* and *Ccr7* all show maximum expressions in the salivary glands of C57BL/6.NOD-*Aec1Aec2* mice at around 16 weeks of age, *Dock1*, *5*, *7* and *8* expressions occur earlier, at around 8 weeks of age. Based on histology, the expressions of *Dock2*, *10*, and *11* are in line with the appearance of T and B cells involved with T cell-mediated cytotoxic responses, while expressions of *Dock1*, *5*, *7* and *8* correspond with our predicted timeframe for physiological and structural changes within the exocrine glands and the apparent appearance of MZB cells. Furthermore, the involvement of Dock8 in B-cell development and function via activation of the BCR signaling molecules CD19 and BTK [65], together with LFA-1-dependent regulation of B and T cell positioning in germinal centers, suggest an environment that promotes IgG antibody responses to T-dependent antigens [66].

## 4. Materials and Methods

### 4.1. Animal Models

C57BL/6J and C57BL/6.NOD-*Aec1Aec2* mice were bred and maintained under specific pathogen free (SPF) conditions within the Department of Pathology’s Mouse Facility with oversight by Animal Care Services at the University of Florida, Gainesville. The C57BL/6.NOD-*Aec1Aec2* mouse is a well-characterized model of primary SS that spontaneously develops all major features of SS in human patients, except lymphomagenesis. The C57BL/6J mouse was used in generating the recombinant inbred C57BL/6.NOD-*Aec1Aec2* mouse [67]. All animals were maintained on a 12-h light-dark schedule and provided food and acidified water ad libitum. Mice (n = 5 per experimental group) were euthanized at either 4, 8, 12, 16 or 20 weeks of age by cervical dislocation after deep anesthetization, the method approved by the Panel on Euthanasia of the American Veterinary Association. There were no indications that this procedure of euthanization affected the preparation of tissues or RNA. Both the breeding and use of these animals for the present studies were approved by the University of Florida Institutional Animal Care and Use Committee (IACUC).

### 4.2. Histology

Salivary glands surgically removed from each mouse at time of euthanasia were placed in 10% phosphate-buffered formalin for 24 h, embedded in paraffin and sectioned at 5 µm thickness. Following deparaffination and dehydration, each section was treated with blocking solution containing donkey serum. Sections were then stained with purified rat anti-mouse CD45R (Clone 30-F11, BD Pharmingen, San Jose, CA, USA) diluted 1:25 and goat polyclonal IgG anti-mouse CD3ε (Clone M-20, Santa Cruz Biotechnology, Santa Cruz, CA, USA) diluted 1:50 in an antibody diluent (Dako, Carpinteria, CA, USA) for 1 h at 25 °C. The slides were then washed with PBS followed by a 1 h incubation with Alexa Fluor 488 donkey anti-goat IgG (H+L) and Alexa Fluor 594 donkey anti-rat IgG (H+L) (Life Technologies, Grand Island, NY, USA). After a thorough wash with PBS, the slides were treated with a Vectashield DAPI-mounting medium (Vector Laboratory, Burlingame, CA, USA) and visualized microscopically at 200×.

### 4.3. RNA Preparations, Microarray Procedures and Microarray Data Analyses

Procedures for the isolation, preparation and quality testing of RNA samples, as well as analyses of microarray data, are described in full detail elsewhere [68,69]. In brief, salivary glands, free of lymph nodes, were excised from C57BL/6.NOD-*Aec1Aec2* and C57BL/6J mice at 4-, 8-, 12-, 16- or 20 weeks of age (n = 5 per group), snap-frozen in liquid nitrogen and stored at −80 °C. Using one lobe from each explanted salivary gland, total RNA from each mouse was individually isolated using RNeasy Mini-Kits (Qiagen, Valencia, CA, USA), tested for purity and quantitated. Each RNA sample (n = 45 total) was hybridized on an Affymetrix 3′ Expression Array GeneChip Mouse Genome 430 2.0 array (Affymetrix, Santa Clara, CA, USA) and annotated (build 32; 06.09.2011). To verify array results, selected gene expressions were carried out using real-time polymerase chain reaction (rt-PCR) analyses. Microarray data were deposited with Expression Omnibus, accession number GSE15640.

Microarray data were normalized using the GCRMA (guanine-cytosine robust multi-array average) algorithm. Normalized data were analyzed using LIMMA (Linear Models for Microarray Analysis), available from the R Development Core Team [70], to identify differential gene expressions. The false discovery rate (FDR) method was used to adjust the *p* values to control for potential false values [71]. These temporal global data represent five equally spaced time points; thus, temporal patterns of gene expressions were used that included linear, quadratic, cubic, and quartic fit regression models. B-statistics were used to determine if a gene showed either positive or negative trends over time. To determine whether the observed number of gene counts exceeded the expected counts, one-tailed *p* values for enrichment of a particular biological process, molecular function, or pathway were calculated using the standard Fisher exact test. Pairwise comparisons between the five groups were made to assess each gene as differentially expressed at 8, 12, 16, and/or 20 weeks of age compared to their 4 weeks expressions. Furthermore, gene set profiling for specific biological processes and/or signal transduction pathways was carried out to match for identical temporal upregulated expressions as a means to support the concept that various individual genes comprise a functional gene set to help compensate for the fact that the encoded proteins may not be equally expressed and/or the RNA:aptamer binding on the arrays may be of different strengths for the represented genes.

## 5. Conclusions

In summary, the temporal transcriptome analyses presented in the current report provide new perspectives into several molecular mechanisms involved in lymphocyte emigrations, presumably of MZB cells, based on the unique time frame of their arrival in the salivary glands of our SS^S^ C57BL/6.NOD-*Aec1Aec2* mice (i.e., the earliest SS disease phase), plus the observed progressive pathophysiologic development of SS/SS-like disease thereafter. In addition, this study clearly shows the power of transcriptomic analyses to establish a foundation for further in-depth cellular investigations of SS-associated bioprocesses that may identify unique novel targets for possible intervention therapies. However, to better define and support the specific signal transduction pathways and their complementary interactive protein sets that culminate in the autoimmune attack against the salivary (and lacrimal) glands in Sjögren’s Syndrome, additional studies using single cells, single cell populations, and unique functional inhibitors must be carried out to confirm the data that are generated with transcriptome studies.

## Figures and Tables

**Figure 1 ijms-22-07176-f001:**
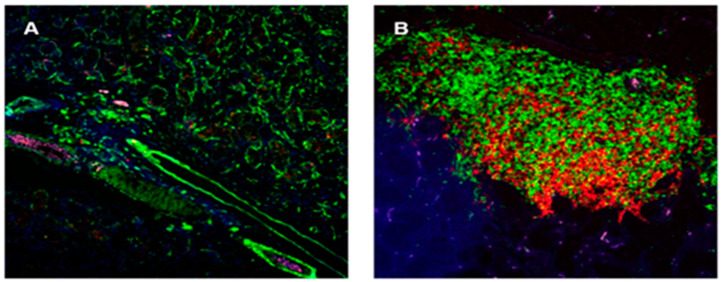
Histological photomicrographs depicting lymphocytic infiltrations present within the salivary glands of C57BL/6.NOD-*Aec1Aec2* mice at 16 (**A**) and 31 (**B**) weeks of age. Infiltrations are age-dependent and contain an ever-changing proportion of both T cells (green fluorescence) and B cells (red fluorescence). Lymphocytic infiltration, especially in SS-susceptible C57BL/6.NOD-*Aec1Aec2* mice, is strongly periductal. Visualization is at 200X magnification.

**Figure 2 ijms-22-07176-f002:**
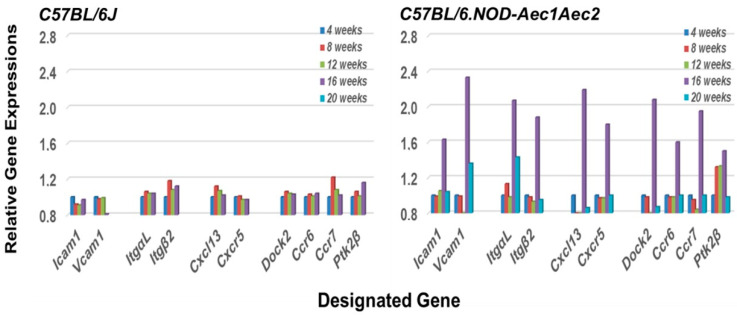
Coordinated temporal activation of factors important for MZB cell emigration from splenic marginal zone (MZ) areas to the salivary glands of C57BL/6.NOD-*Aec1Aec2* mice at onset of SS-like disease. Transcriptomic data reveal that genes encoding ICAM1 and VCAM1 (Receptors for LFA1), LFA-1 (*ItgaL and Itgβ2*), *Cxcl13*, *Cxcr5* (the Cxcl13 receptor), *Dock2*, *Ccr6*, *Ccr7*, and *Ptk2β* are all upregulated in the salivary glands of C57BL/6.NOD-*Aec1Aec2* mice at 16 weeks of age (**right panel**), the approximate time point when the immune response is transitioning to the adaptive immune response phase. In contrast, no similar upregulated expressions of these genes are seen in the salivary glands of C57BL/6J mice (**left panel**).

**Figure 3 ijms-22-07176-f003:**
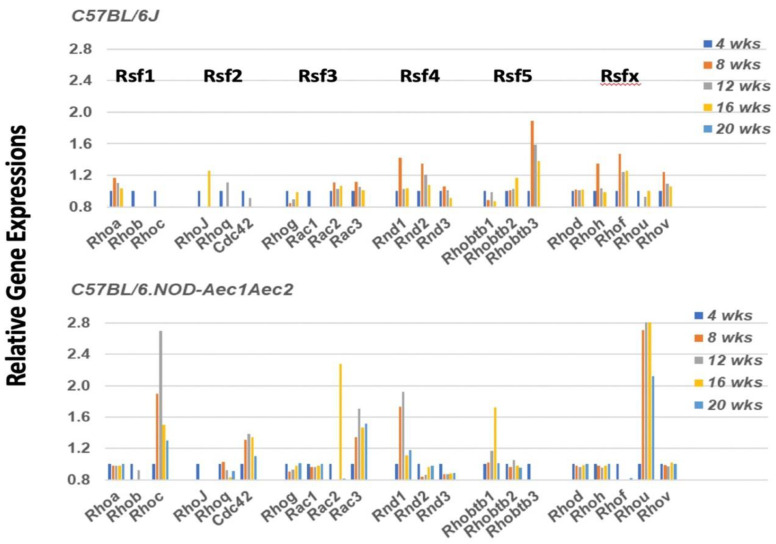
Unique upregulation profile of genes encoding for members of the Rho-GTPase family of proteins in the salivary glands of C57BL/6.NOD-*Aec1Aec2* mice. Rho-GTPase family proteins are important in regulating the cellular homeostasis of the GTP<>GDP system. The Rho-GTPase family consists of 21 members, of which 16 are divided among 5 subfamilies (Rsf1, Rsf2, Rsf3, Rsf4 and Rsf5), with 5 unassigned (designated Rsfx here). For cells in the non-activated state, Rho-GTP molecules are mostly associated with cellular membranes, but during cell activation the molecules dissociate to the cytoplasm by an as-yet unknown mechanism. In the salivary glands of C57BL/6.NOD-*Aec1Aec2* mice, genes encoding for *Rhoc*, *Rhou*, *Cdc42*, *Rac3*, and *Rnd1* are upregulated starting at 8 weeks of age, with *Rac2* and *Rhobtb1* upregulated starting at 16 weeks of age (**lower panel**). In contrast, the salivary glands of C57BL/6J mice showed a weak upregulated expression for *Rnd1*, *Rnd2*, *Rhobtb3*, *Rhoh*, *Rhof* and *Rhov* at 8 weeks of age, but this was not prolonged.

**Figure 4 ijms-22-07176-f004:**
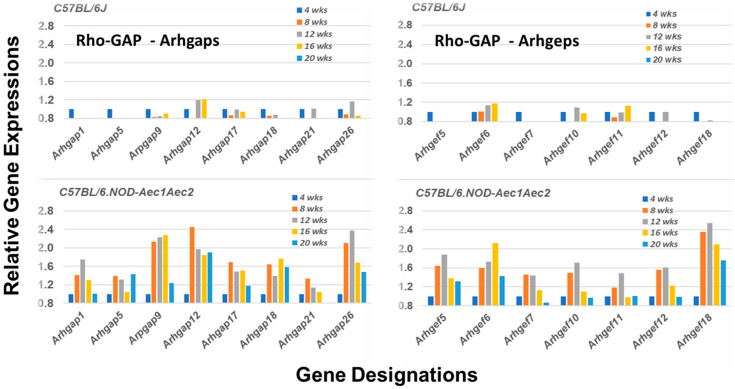
Unique upregulation profiles for genes encoding for members of the Rho-GTPase GAP and GEF families of proteins. Transcriptome data showing the temporal expressions of 8 of the 23 *Arhgap* genes and 7 of the 17 *Arhgef* genes upregulated in the salivary glands of C57BL/6.NOD-*Aec1Aec2* mice (**lower panels**). In each case, upregulated expressions started around 8 weeks of age. In contrast, expressions of these *Arhgap* and Arhgef genes remain downregulated.

**Figure 5 ijms-22-07176-f005:**
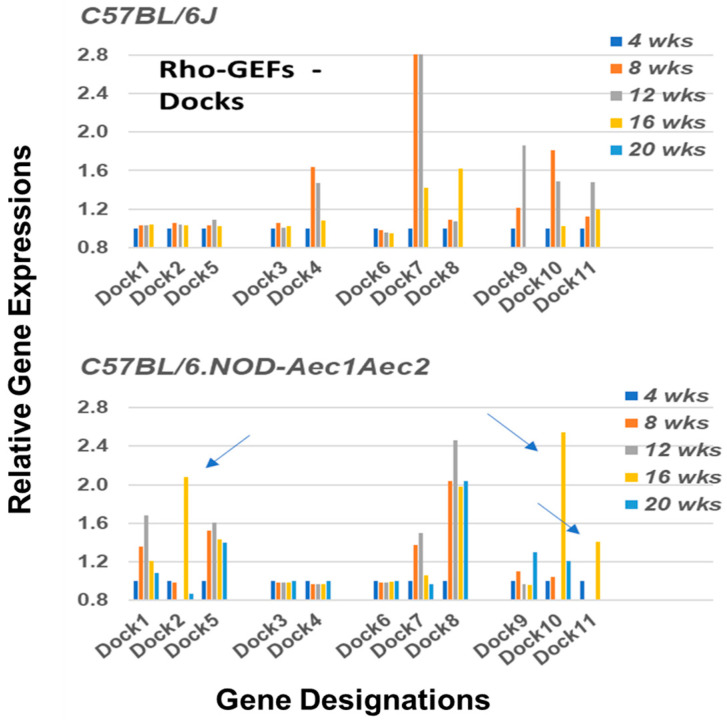
Upregulated profiles of genes encoding for members of the Rho-GTPase Dock family of proteins. Transcriptome data showing the temporal expressions of the *Dock* genes upregulated in the salivary glands of C57BL/6.NOD-*Aec1Aec2* mice (**lower panel**) versus C57BL/6J mice (**upper panel**). In the SS-susceptible C57BL/6.NOD-*Aec1Aec2* mice three genes, *Dock2*, *Dock10*, and *Dock11*, reported to be uniquely associated with B and T lymphocyte functions, exhibit a concomitant upregulated short-term expression at 16 weeks of age (arrows), while *Dock1*, *5*, *7* and *8* are each upregulated starting at 4 weeks of age and remain activated out to 16 or 20 weeks of age. In contrast, in the SS-non-susceptible C57BL/6J mice, *Dock4*, *7*, *9*, *10* and *11* exhibit an upregulated expression starting at 4 weeks of age, while *Dock8* shows a unique expression pattern being upregulated at 16 weeks of age.

**Figure 6 ijms-22-07176-f006:**
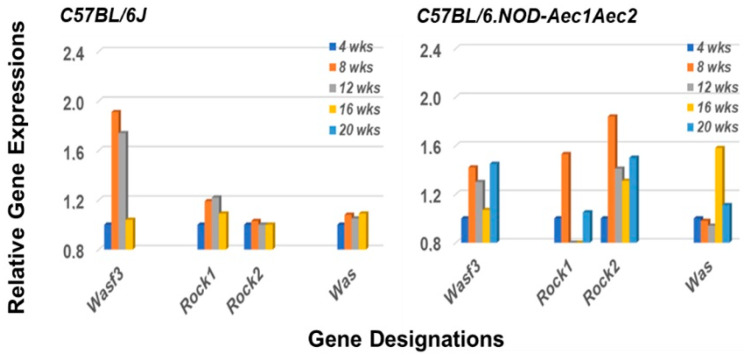
Rho-Arhgef pathway activations. Rho-Arhgef molecules are associated with several signal transduction pathways including Rac-GTP, Rho-GTP and CDC42-GP, each leading to downstream activation of a biological process important for cellular migrations. These three pathways interact, respectively, with Wasf, Rock, and Was protein family members. In the salivary glands of C57BL/6.NOD-*Aec1Aec2* mouse these appear to be Wasf3, Rock1, Rock2 and Was (right panel). The genes encoding *Wasf3*, *Rock1* and *Rock2* show an activation starting at 4 weeks of age, while the gene encoding *Was* is upregulated uniquely at 16 weeks of age. In the salivary glands of C57BL/6J mice, a strong activation is observed with *Wasf3*, and possibly a weak response by *Rock1*.

**Figure 7 ijms-22-07176-f007:**
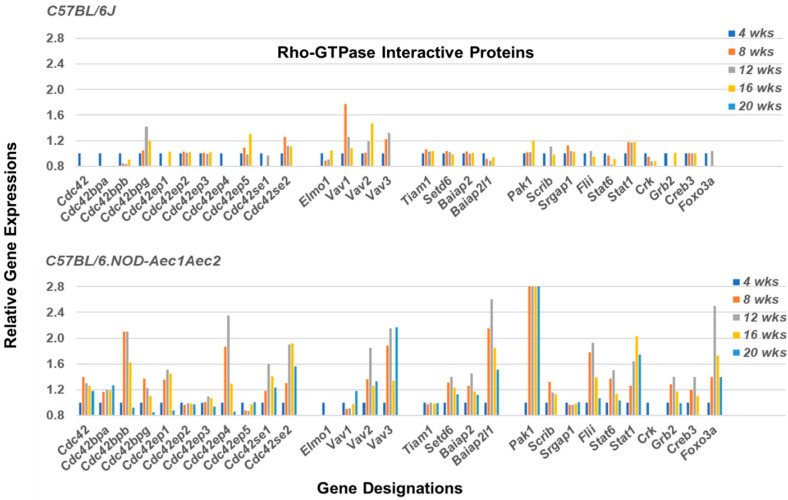
Temporal and differential gene expressions of various factors involved with signal transduction pathways regulated by members of the Rho-GTPase families of proteins. Transcriptome data of multiple genes (i.e., *Cdc42*, *Vav*, and *Baiap* (brain-specific angiogenesis inhibitor -1 associated protein) family members, plus *Scrib1*, *Setd6*, *Grb2*, *Flii*, *Creb3* and *Foxo3a*) exhibit upregulated expressions in the salivary glands of C57BL/6.NOD-*Aec1Aec2* mice, virtually all starting at 8 weeks of age (**lower panel**). C57BL/6J mice exhibit a general lack of upregulated expression, except for *Vav1* and possibly *Cdc42bpg* and *Vav2* (**upper panel**).

## Data Availability

Microarray data are publicly available and deposited with The Gene Expression Omnibus, Accession Numbers GSE15640 and GSE36378.

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
