# Peer review of "Upregulated Chemokine and Rho-GTPase Genes Define Immune Cell Emigration into Salivary Glands of Sjögren’s Syndrome-Susceptible C57BL/6.NOD-Aec1Aec2 Mice"

_ijms, 2021, doi:10.3390/ijms22137176_

Round 1
Reviewer 1 Report
In this study, Peck et al performed bulk RNA-seq transcriptomic analysis of salivary glands in wild type mice and a mouse model of Sjogren's Syndrome (SS) (C57BL/6.IL14α transgenic mouse) and demonstrated the temporal gene expression changes that occur with the development of SS. They show demonstrate specific gene signatures with upregulation of certain integrins, chemokines, and signal transduction pathways (Rho-GTPase family) associated with the pathogenesis of SS with time. This an interesting, very detailed, exploratory and hypothesis-generating study that may help reveal novel mechanistic insights in SS. I have the following critiques and recommendations:
1. Results/Discussion: The authors speculate that the upregulation of certain integrins were from marginal zone B cells (MZB). However, given that their data was bulk-RNA seq, one cannot make this conclusion from bulk tissue without single cell analysis. Have the authors isolated MZB from salivary gland biopsies and performed gene expression analyses solely on this immune subset?
2. Results/Discussion: Figure shows IHC of salivary glands in wild type vs SS mice models delineating T and B cell infiltration. Have the B and T cell immunophenotypes (e.g. CD4, CD8, Th17, naive vs memory B cells, MZB) and proportions in salivary glands been characterized?
3. Results: Figures 2-7 should show which findings are statistically significant.
4. Methods/Results: Are there any differences in B cell (or MZB) gene expression from B cells in peripheral blood versus lymph node vs salivary glands of this mice model of SS? It would be interesting to determine any tissue-specific changes as B cells trafficking from blood or lymphoid tissue (e.g. lymph nodes) to salivary glands.
5. Discussion: The authors should discuss whether similar gene expression patterns have been described from salivary gland biopsies in patients with Sjogren's syndrome.
6. Discussion: "Thus, we propose that this early disease stage can be divided into a ¨passive¨ emigration involving MZB cells and an ¨active¨ emigration involving recruitment of additional leukocyte populations." These conclusions are based on gene expression alone with any functional assays. Have prior studies evaluating leukocyte trafficking of specific T and B cells subsets (e.g. MZB) to salivary glands been performed in these mice model of SS?
7. Discussion: Blockade of gut tropic integrins (e.g. vedolizumab, etrolizumab) have been shown to have some therapeutic efficacy in patients with inflammatory bowel disease. Given that T and B cell trafficking and infiltration play also play a role in the pathogenesis of SS, have there been any studies evaluating blockade of leukocyte trafficking as a therapeutic approach in murine SS?
8. Discussion: Is anything known about the B cell repertoire (BCR)/immunoglobulin repertoire of MZB in this mouse model of SS? Are the B-cell mediated mechanisms of SS known? Anti-Ro and Anti-La antibodies are sometimes used as biomarkers from SS in human patients. It would be interesting to understand if MZB in salivary glands produced auto-reactive antibodies.
Reviewer 2 Report
The manuscript is interesting and well written. However, I suggest to briefly discuss and add ad reference paper by Negrini te al concerning S.Sjogren published in june 2021
